# Association of primary postpartum hemorrhage with inter-pregnancy interval in urban South Ethiopia: A matched nested case-control study

Belayneh Hamdela Jena[1,2]*, Gashaw Andargie Biks[3], Yigzaw Kebede Gete[1], Kassahun Alemu Gelaye[1]

1 Department of Epidemiology and Biostatistics, Institute of Public Health, College of Medicine and Health Sciences, University of Gondar, Gondar, Ethiopia, 2 Department of Public Health, College of Medicine and Health Sciences, Wachemo University, Hossana, Ethiopia, 3 Department of Health System and Policy, Institute of Public Health, College of Medicine and Health Sciences, University of Gondar, Gondar, Ethiopia

* bhamdela@gmail.com

**Data Availability Statement:** All relevant data are within the paper and its Supporting information files.

## Abstract

### Background

Globally, postpartum hemorrhage is the leading preventable cause of maternal mortality. To decrease postpartum hemorrhage-related maternal mortalities, identifying its risk factors is crucial to suggest interventions. In this regard, little is known about the link between primary postpartum hemorrhage and inter-pregnancy interval in Ethiopia, where more than half of pregnancies occur shortly after the preceding childbirth. Therefore, we aimed to elucidate the association of primary postpartum hemorrhage with an inter-pregnancy interval in urban South Ethiopia.

### Methods

A community-based matched nested case-control study was conducted among a cohort of 2548 pregnant women. All women with primary postpartum hemorrhage during the follow-up (n = 73) were taken as cases. Women who were randomly selected from those without primary postpartum hemorrhage (n = 292) were taken as controls. Cases were individually matched with controls (1:4 ratio) for age group and location. A conditional logistic regression analysis was done using R version 4.0.5 software. Statistically, a significant association was declared using 95% CI and p-value. Attributable fraction (AF) and population attributable fraction (PAF) were used to estimate the public health impacts of the inter-pregnancy interval.

### Results

This study found out that more than half (66%) of primary postpartum hemorrhage was attributed to inter-pregnancy interval <24 months (AF = 66.3%, 95% CI: 37.5, 82.5%). This could be prevented if the inter-pregnancy interval was increased to 24–60 months. Likewise,

**Funding:** The authors received no specific funding for this work.

**Competing interests:** The authors have declared that no competing interests exist.

nearly half (49%) of primary postpartum hemorrhage in the study population could be prevented if the inter-pregnancy interval <24 months was prevented. Additionally, primary postpartum hemorrhage was attributed to antepartum hemorrhage, prolonged labour and multiple pregnancies.

## Conclusions

Primary postpartum hemorrhage was associated with inter-pregnancy interval under 24 months, highlighting the need to improve postpartum modern contraceptive utilization in the community. Counseling couples about how long to wait until subsequent pregnancy and the risk when the inter-pregnancy interval is short need to be underlined.

## Introduction

Primary postpartum hemorrhage (PPH) is generally defined as a loss of 500 ml or more amounts of blood within 24 hours after vaginal birth or 1000 ml or more blood following cesarean delivery [1]. It is a common form of PPH, as more bleeding occurs within this time frame after delivery. Bleeding during delivery in small amount is a natural phenomenon. But, it can be massive life-threatening condition and causes maternal death when the bleeding is profuse or ongoing and cause changes in vital signs [2].

Globally, PPH is recognized as the leading preventable cause of maternal mortality, which accounts for nearly one quarter (25%) of all maternal deaths annually [1–4]. PPH accounts for 8% of maternal deaths in developed countries and 20% of maternal deaths in developing countries [5]. PPH affects about 18% of all deliveries [6]. Specifically, primary PPH is a common one, contributing to 19.7% of all pregnancy-related maternal deaths worldwide [7]. Those mothers who survived from death often face serious complications following immediate PPH such as shock, myocardial ischemia, and anemia that has long-term clinical effects on the women and subsequent births [8, 9]. These acute and chronic complications of PPH cause considerable suffering for mothers who are the victims of bleeding and their families, especially the newborn who needs immediate care and support from the mother such as breast-feeding and good temperature from skin-to-skin contact. PPH disturbs these crucial cares for the newborn because the mother herself needs life-saving care during the condition. Moreover, the complications cause a heavy burden on health systems in general [10]. Low and middle-income continents such as Africa and Asia take the highest burden of PPH-related morbidities and mortalities due to inadequate access and unavailability of quality obstetric services including lack of surgical management of atony in these settings [11]. However, recent publications have reported that the rate and related maternal morbidities were increasing in high-income countries such as Canada (6.2%) [8], the United States (2.9%) [12], United Kingdom (13.8%) [13], Japan (8.7%) [14] and Ireland (4.1%) [15]. Prevalence of postpartum hemorrhage was 26% in Africa, 13% in North America and Europe, and 8% in Latin America and Asia [16]. The prevalence of primary PPH in Ethiopia ranges from 5.8% in Dessie [17] and 7.6% in Debre Tabor [18] to 16.6% in the South region [7].

Previously conducted studies identify factors that put mothers at a higher risk of developing PPH such as previous history of PPH [19], age $\geq$ 35 years [20, 21], uterine over-distention (due to multiple pregnancy [20, 22] and fetal macrosomia [20, 23]), multiparty [17, 21, 24], prolonged labour [17, 24], induced or augmented labour [24, 25], pre-eclampsia [22], chorioamnionitis [26], caesarean section [20, 25] and episiotomy [2, 23]. Inter-pregnancy interval

(IPI) (a time elapsed from live birth to subsequent conception or woman's last menstrual period) <18 months [27] and <24 months [28] were reported to increase the risk of primary PPH. However, a recent study from Tanzania has shown that IPI <24 months has no effect on primary PPH [29]. It was also noted that about 20% of PPH cases could present without known risk factors [2, 7].

The exact mechanism that how short IPI results in adverse maternal outcomes such as PPH remain unclear. However, it is hypothesized based on the theory of maternal depletion, which suggests that short intervals between pregnancies do not allow the mother to recover from the abnormal conditions during the preceding pregnancy and childbirth, such as abnormal process of remodeling of the endometrial vessels, incomplete healing of uterine scar that further leads to utero-placental bleeding disorders such as retained placenta, uterine atony, and uterine rupture [29–31].

In Ethiopia, maternal mortality remains one of the highest in the world, 412 per 100,000 live births [32]. Hemorrhage, specifically, PPH is the main direct cause of maternal death [33–35]. Ethiopia has to go a long journey to achieve the target set to reduce maternal mortality to below 70 per 100,000 live births or not exceed its double (140 per 100,000 live births) by 2030 [35]. The maternal mortality ratio is still highest and can be difficult to reduce to the desired level (70 per 100,000 live births) unless PPH is substantially reduced. Moreover, there is a high rate of fertility and half of the second and higher pregnancies occur to women within a short duration after preceding childbirth [36]. However, whether the short interval (IPI<24 months) between pregnancies has an effect on primary PPH or not is understudied. Available studies (aforementioned) reported inconsistent results. Our study was supposed to supplement those results. Additionally, the world health organization (WHO) called for further research on the effect of IPI on adverse pregnancy outcomes, including primary PPH [37].

Based on these rationales, we hypothesized that IPI<24 months increases the risk of primary PPH as compared to IPI 24–60 months. Therefore, we aimed to elucidate the association between primary PPH and IPI in urban South Ethiopia. The finding will contribute to reducing IPI-related PPH and subsequent risk of maternal death by improving maternal health services at local and national levels.

## Methods

### Study design and setting

A community-based matched nested case-control study design was applied to this study. It is a part of a community-based prospective cohort study that was conducted among pregnant women in five urban settings (Hossana, Shone, Gimbichu, Jajura and Homecho) in the Hadiya zone, South Ethiopia. Hadiya zone is one of the zones in southern nations, nationalities and peoples region in Ethiopia. The zone is located 232 km far from the capital city, Addis Ababa, and 294 km far from its regional capital city, Hawassa. In the zone, there are 1 general hospital, 3 primary hospitals, 62 health centers and 311 health posts that offer health services for the community (Hadiya Zone Health Bureau report-unpublished).

### Participants

For this study, a cohort of pregnant women were enrolled at the end of 1st trimester of confirmed pregnancy (after 12 weeks of gestation) via house-to-house identification and registration every three months, for a total of nine months. An enrolment was done from July 08/2019 to March 30/2020 by trained midwives. During the recruitment, study participants were included in the study based on the eligibility criteria for the main exposure variable (inter-pregnancy interval). The inclusion criteria were women who: were pregnant at the time of

recruitment, had a live birth during the most recent childbirth, and were able to recall the date of last childbirth. Women who had a recent stillbirth, a recent abortion, and those who did not show a willingness to be followed were excluded.

A total of 2578 pregnant women were enrolled and the enrolled pregnant women were followed until September 30/2020. Of 2578, 1273 were pregnant women with IPI <24 months (exposed group) and 1305 were those with IPI 24–60 months (unexposed group), based on World health organization (WHO) recommendation for pregnancy spacing [37]. For this particular study, all primary PPH cases occurring during a follow-up time (July 08/2019 to September 30/2020) were taken as cases. The controls were women delivering without primary PPH. A 1:4 ratio was used because cases (primary PPH) were relatively small in size. Hence to increase the statistical power, four controls were selected for each case. For every primary PPH case, four controls were randomly selected from the frame of the cohort (risk set) by using a random numbers generator in open-Epi software [38]. We matched a case with four controls each for age group and location or kebele (the lowest administrative unit in Ethiopia). The individual matching approach was applied to pair cases with controls. Age is categorized in five years intervals (as age is subjective and less accurately reported in most people in Ethiopia due to the absence of a vital registration system so far). Location (kebele) was easily identifiable (objective) so that simply matched without recoding into intervals or groups. Thus, individual matching possibly makes the cases to be very similar to controls. Matching in location (kebele) also makes the cases and controls very similar for many unobserved sources of heterogeneity. For example, it makes the distance to the health facility, the health facility that they receive maternal health services (such as antenatal, delivery, and postnatal care), social, cultural, and other neighborhood circumstances. Thus, reduce geographic disparities as this study was conducted in multiple (five) locations (Fig 1).

## Variables and definitions

**Outcome variable.** The dependent/outcome variable was primary PPH (a hemorrhage that occurred within 24 hours after delivery of a child).

**Exposure variable.** The main exposure variable was inter-pregnancy interval (a time elapsed from live birth to subsequent conception or woman's last menstrual period) [37].

**Covariate/Confounding variables.** Covariate/confounding variables were: 1) Socio-demographic and economic variables such as maternal age, marital status, education, occupation, family size and wealth status. 2) Reproductive and maternal health service-related characteristics such as parity, contraceptive use, number of antenatal care visits, type of pregnancy, antepartum hemorrhage (APH), pre-eclampsia/eclampsia, the progress of labour, prolonged/obstructed labour, mode of delivery, receiving oxytocin and weight of the new-born. Definitions for some technical terms are as follow:

Parity: is the number of times a woman gives birth, irrespective of the birth outcome.

Antepartum hemorrhage: is vaginal bleeding after 28 weeks of gestation.

Pre-eclampsia: is pregnancy-related high blood pressure (140/90 mmHg) and protein in the urine.

Eclampsia: is when a woman with pre-eclampsia develops convulsion or comma.

Prolonged labour: is a duration of true labour that exceeds 12 hours, irrespective of the stages.

Type of pregnancy: refers to the number of babies born at the time of delivery (single or multiple).

Mode of delivery: is whether a woman gives birth spontaneous vaginally, assisted vaginally with instruments (forceps and vacuum) or via cesarean section.

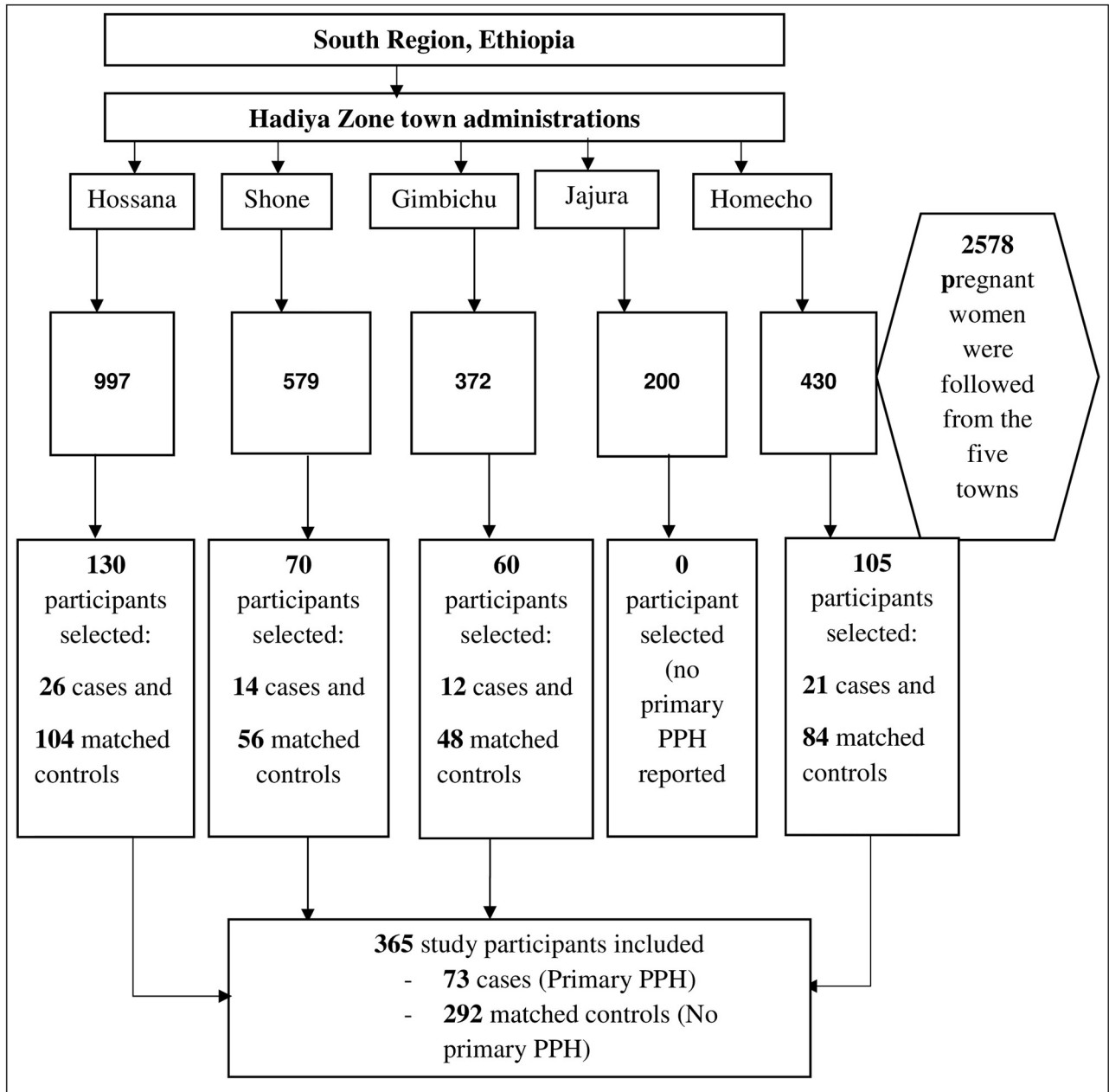

**Fig 1. Schematic presentation of sampling procedure for the study of primary PPH in the urban South Ethiopia, July 2019-September 2020.**

Progress of labour: mean whether labour was going on its natural course or assisted via augmentation or induction.

Birth weight: the weight of the new-born at birth measures in grams.

## Data sources

Before baseline data collection, the questionnaire was prepared from existing related literature (published articles and Ethiopia Demographic and Health Surveys) based on the study objectives [7, 27, 36, 39]. English version was translated to Amharic version by two native speakers

of Amharic language (one was public health and the other was English language and literature in professions). Then back translation to English was done by another two individuals who could speak English (again one was from public health and the other from English language and literature). The questionnaire was pre-tested on 50 pregnant women in Durame town where the actual study population is culturally related. The investigators have amended the pre-test.

Baseline data about sociodemographic and reproductive variables including the main exposure variable (IPI) were collected at the household level during enrolment via face-to-face interviews. Outcome (primary PPH) and other clinical data such as pre-eclampsia, prolonged labour, the progress of labour, mode of delivery, and weight of baby were collected from clients' charts, during the time of delivery before discharge made. Ten trained midwives collected data and five public health professionals made supervisions. The data collectors at each health facility were assigned and the list of participants was given for each of them. The birth attendants were informed to consult a nearby gynaecologist in a zonal hospital and emergency medicine specialist in district primary hospitals to make the diagnosis of PPH in addition to their expertise.

## Measurements

**Outcome ascertainment.** There is no standardized tool that quantifies the amount of blood loss that works in all setups to diagnose PPH. Visual estimation is one of the commonly used methods and is also used in our health facilities supplemented with vital signs such as the drop in blood pressure, 10% reduction in hemoglobin from prepartum and increase in heart rate (tachycardia). Signs of hypovolemic shock and cases with blood transfusion were also used to diagnose PPH in addition to visual estimation. For this study, in particular, we used clinically diagnosed PPH based on the combination of those methods and taken from clients' charts. Then primary PPH was categorized as a binary variable (0 = absent, 1 = present).

**Exposures ascertainment.** The exposure variable (IPI) was ascertained by asking women about the date of most recent childbirth and the last menstrual period. IPI was computed by subtracting the date of recent childbirth from the date of the last menstrual period (LMP). For women who had difficulty in recalling the date of LMP, Ultrasound was used to estimate gestational age. LMP was computed by subtracting the duration of gestation, and then the value of IPI was calculated [37]. To be in line with the WHO recommendation, women with IPI <24 months were categorized as exposed group and IPI 24–60 months as unexposed group.

**Covariate/Confounding ascertainment.** Potential confounding variables (such as age, parity, education) are those variables that have an association with an exposure (IPI) and an outcome (primary PPH). Covariates were those factors that have an association with the outcome (primary PPH) but not with the exposure (IPI). These covariates/confounders were identified by prior theoretical knowledge and literature [17, 20, 21, 24, 40]. The covariates/confounders were measured according to their definitions mentioned above (variables and definition section) and as follows: Antepartum hemorrhage assessed for recent pregnancy in two ways: firstly from the patient chart if data is available. Secondly, by interviewing the woman if she had any vaginal bleeding after seven months of gestation (after 28 weeks). If at least one data was obtained from the two sources, she would have been diagnosed as having APH (0 = absent, 1 = present). Pre-eclampsia/eclampsia, prolonged labour, management of the third stage of labour, and other clinical data were taken from clients' charts and categorized as 0 = absent, 1 = present.

Wealth index was measured using household assets for urban residence, which consists of the following items: owner of the house, number of rooms, the material of the roof, material of

the floor, material of the exterior wall, source of drinking water, type of latrine, type of cooking materials (1 = electricity, 0 = wood/charcoal/biogas/natural gas, etc.), source of income, and presence or absence of; cell phone, refrigerator, radio, television, stove, chair, table, watch, modern bed, bicycle, Bajaj, motorcycle, car, donkey/horse cart, and bank account. Each item was categorized into two (1 = yes and 0 = no). Latrine and water sources were categorized as an improved and unimproved facility based on the world food program and WHO recommendations. Principal component analysis was done to generate the components. Finally, the ranking was done into three quintiles (low, middle and high).

## Sample size

In nested case-control studies, the size of the sample is often dependant on the number of observed cases from a cohort study and their corresponding controls. Therefore, for this matched case-control study, a sample size of 365 (73 cases and 292 matched controls) was considered.

## Analysis

Data was entered in Epi-data version 3.1 software [41] and exported to R version 4.0.5 software [42] for the analysis. Frequencies and percentages using cross-tabulation were calculated for categorical variables and discreet continuous variables. For missing data a complete case analysis approach was applied. A conditional binary logistic regression analysis for matched data was fitted (a case matched and paired with 4 controls for the same age group and location). All independent variables which have shown significant association with primary PPH in binary conditional logistic regression model at $P<0.25$ were included in the multivariable conditional logistic regression model. Then statistically significant association was declared at $P<0.05$ and confidence intervals for odds ratio that did not include 1. The results were presented using odds ratio and interpreted using attributable fraction. Attributable fraction (AF) was calculated from the adjusted odds ratio to estimate the impact of public health of the exposures (IPI).

## Quality control measures

Data collectors and supervisors were trained for two days on the concept of the questionnaire and how to approach the participants ethically. A pre-test was done in a similar setting (Durame town). Supervisors checked the data collection process closely. To minimize recall bias, we limited the date of recent childbirth in the last five years to help women in recalling data of last childbirth. Family members such as husband, grandparent, and mother-in-law were also involved to recall the date of childbirth. To minimize selection bias, we used community-based recruitment to include pregnant women during the study period using predefined eligibility criteria. Additionally, Ultrasound was used for those women who had difficulty in remembering the date of LMP due to different reasons such as contraceptive use and breastfeeding. A sensitivity analysis was done to check the impact of misclassification (if it exists) of exposure (IPI), as it depends on the ability of women in recalling the dates of last childbirth and last menstrual period. Epi-data was used to control errors during data entry. Data were explored to check outliers and missing values. We applied matched and paired analysis (conditional logistic regression) which also helps to reduce bias due to confounding in addition to multivariable analysis.

## Ethics approval and consent to participate

Ethical clearance was obtained from the institutional review board (IRB) of the University of Gondar, with registration number: O/V/P/RCS/05/1051/2019. Permission was obtained from

regional and local health offices. Study participants were informed about how they were included in the study, the purpose of the study, their rights to withdraw or continue, and potential benefits and harms of the study. A written informed consent form was prepared and attached with the questionnaire to obtain approval from each study participant by signature or fingerprint.

## Results

### Cohort information

During the study period, a follow-up was made for 2578 pregnant women. Of these, 29 (1.14%) were lost of follow-up (21 due to end of the study period, 8 no information at all including via phone calling) and their pregnancy outcomes could not be ascertained. Of 29 lost of follow-up, 14 were from exposed and 15 from unexposed groups. The pregnancy outcomes were ascertained for 2549 study participants. At the end of a follow-up, 73 (2.9%) women had primary PPH (S1 Fig).

### Socio-demographic and economic information

The mean age of both cases and controls was 27.1 years ± 3.1 and 27.5 years ± 3.5 respectively. More controls 231 (79.1%) were attended formal education (1–12 grade or above) than the cases 55 (75.3%) (Table 1).

**Table 1. Socio-demographic and economic variables of the participants in urban South Ethiopia, July 2019-September 2020.**

| Variables | Cases = 73 | Matched controls = 292 |
|---|---|---|
| | n (%) | n (%) |
| **Religion** | | |
| Protestant | 68 (93.2) | 264 (90.4) |
| Orthodox | 2 (2.7) | 13 (4.5) |
| Catholic | 1 (1.4) | 6 (2.1) |
| Muslim/Apoplectics | 2 (2.7) | 9 (3.1) |
| **Ethnicity** | | |
| Hadiya | 70 (95.9) | 270 (92.5) |
| Kembata/ Siltie/Guragie | 3 (4.1) | 22 (7.5) |
| **Maternal educational status** | | |
| No formal education | 18 (24.7) | 61 (20.9) |
| Attended formal education | 55 (75.3) | 231 (79.1) |
| **Maternal occupation** | | |
| Employed | 5 (6.8) | 26 (8.9) |
| Housewife | 58 (79.5) | 210 (71.9) |
| Merchant/vender | 10 (13.7) | 56 (19.2) |
| **Family size (n = 363)** | | |
| < 5 | 59 (80.8) | 230 (78.8) |
| ≥ 5 | 14 (19.2) | 62 (21.2) |
| **Wealth status** | | |
| Low | 29 (39.7) | 116 (39.9) |
| Middle | 10 (13.7) | 63 (21.6) |
| High | 34 (46.6) | 112 (38.5) |

## Reproductive and maternal health service-related characteristics

The mean age of women at first childbirth for cases and controls was 21.9 ± 2.9 years and 21.3 ± 2.8 years respectively. For current pregnancy, more proportion of controls 264 (90.4%) had given birth via spontaneous vaginal mode as compared to the cases 62 (84.9%). More proportion of cases had given birth via cesarean section 4 (5.5%) and instrumental delivery 7 (9.6%) than controls (Table 2).

**Table 2. Reproductive and maternal health service-related characteristics of the participants in urban South Ethiopia, July 2019-September 2020.**

| Variables | Cases = 73 | Matched controls = 292 |
|---|---|---|
| | n (%) | n (%) |
| **Inter-pregnancy interval in months** | | |
| <24 | 54 (74) | 142 (48.6) |
| 24–60 | 19 (26) | 150 (51.4) |
| **Modern contraceptive use after the recent childbirth (n = 363)** | | |
| Used | 34 (46.6) | 131 (45.2) |
| Not used | 39 (53.4) | 159 (54.8) |
| **Pre-eclampsia/eclampsia** | | |
| Present | 3 (4.1) | 7 (2.4) |
| Absent | 70 (95.9) | 285 (97.6) |
| **Progress of labour** | | |
| Unassisted/normally progressed | 66 (90.4) | 260 (89) |
| Augmented/induced | 7 (9.6) | 32 (11) |
| **Parity** | | |
| 1 | 31 (42.5) | 99 (33.9) |
| 2 | 19 (26) | 89 (30.5) |
| ≥ 3 | 23 (35.1) | 104 (35.6) |
| **Birth weight** | | |
| ≤ 4000gm | 64 (87.7) | 284 (97.3) |
| > 4000gm | 9 (12.3) | 8 (2.7) |
| **Mode of delivery** | | |
| Spontaneous vaginally | 62 (84.9) | 264 (90.4) |
| Cesarean section | 4 (5.5) | 11 (3.8) |
| Instrumental | 7 (9.6) | 17 (5.8) |
| **Antepartum hemorrhage** | | |
| Present | 11 (15.1) | 7 (2.4) |
| Absent | 62 (84.9) | 285 (97.6) |
| **Number of antenatal care visits (n = 363)** | | |
| <4 | 39 (54.2) | 196 (67.4) |
| ≥ 4 | 33 (45.8) | 95 (32.6) |
| **Prolonged labour** | | |
| Present | 20 (27.4) | 20 (6.8) |
| Absent | 53 (72.6) | 272 (93.2) |
| **Type of pregnancy** | | |
| Singleton | 65 (89) | 285 (97.6) |
| Multiple (twins) | 8 (11) | 7 (2.4) |
| **Oxytocin during 3rd stage of labour** | | |
| Given | 69 (94.5) | 268 (91.8) |
| Not given | 4 (5.5) | 24 (8.2) |

## Association of primary postpartum hemorrhage with inter-pregnancy interval

In the binary conditional logistic regression model, eight variables: IPI, parity, mode of delivery, APH, prolonged labour, number of ANC visits, the weight of baby, and type of pregnancy were associated with primary PPH at P<0.25. When these variables fitted in the multivariable conditional logistic regression model, four variables: IPI, APH, prolonged labour, and type of pregnancy were found to be associated with primary PPH with 95% CI at P<0.05.

In this study, women who had a pregnancy within 24 months after the preceding childbirth were nearly three times (AOR = 2.97, 95% CI: 1.6, 5.7) more likely to experience primary PPH as compared to those who had 24–60 months intervals. This means about 66% of primary PPH was attributed to IPI <24 months as compared to 24–60 months (AF = 66.3%, 95% CI: 37.5, 82.5%) (Table 3).

**Table 3. Multivariable conditional logistic regression model for the association of primary postpartum hemorrhage with the inter-pregnancy interval in urban South Ethiopia, July 2019-September 2020.**

| Variables | Crude OR (95%CI) | Adjusted OR (95%CI) | AF (95%CI) |
|---|---|---|---|
| **Inter-pregnancy interval in months** | | | |
| <24 | 2.99 (1.7, 5.3)*** | 2.97 (1.6, 5.7)*** | 66.3% (37.5, 82.5) |
| 24–60 | 1 | 1 | 1 |
| **Parity** | | | |
| 1 | 1 | 1 | |
| 2 | 0.51 (0.23, 1.1)• | 0.56 (0.21, 1.5) | |
| ≥ 3 | 0.35 (0.11, 1.1)• | 0.46 (0.10, 2.1) | |
| **Number of antenatal care visits** | | | |
| <4 | 1 | 1 | |
| ≥ 4 | 1.9 (1.1, 3.4)* | 1.8 (0.92, 3.46) | |
| **Antepartum hemorrhage** | | | |
| Present | 6.9 (2.6, 18.9)*** | 4.9 (1.5, 16.2)** | 79.6% (33.3, 93.8) |
| Absent | 1 | 1 | 1 |
| **Type of pregnancy** | | | |
| Singleton | 1 | 1 | 77.8% (23.1, 93.5) |
| Multiple (twins) | 4.6 (1.7, 12.6)** | 4.5 (1.3, 15.3)* | 1 |
| **Prolonged labour** | | | |
| Present | 4.8 (2.4, 9.5)*** | 3.2 (1.2, 8.5)* | 68.8% (16.7, 88.2) |
| Absent | 1 | 1 | 1 |
| **Mode of delivery** | | | |
| Spontaneous vaginally | 1 | 1 | |
| Cesarean section | 1.5 (0.46, 5.0) | 0.87 (0.19, 3.8) | |
| Instrumental | 1.8(0.69, 4.5)• | 0.82 (0.24, 2.8) | |
| **Birth weight** | | | |
| ≤ 4000gm | 1 | 1 | |
| > 4000gm | 4.9 (1.8, 13.2)** | 3.3 (0.87, 12.8) | |

Keys: Significant

*** = P<0.001,

** = P<0.01,

* = P<0.05.

• = P<0.25

OR: Odds Ratio. CI: Confidence Interval. 1 = reference category. AF: Attributable Fraction.

### Sensitivity analysis

We conducted a sensitivity analysis to whether the cutoff 24 months resulted in misclassification of exposure (IPI) by decreasing and increasing 1 month as follow: firstly, when IPI cutoff decreased by 1 month (IPI<23 vs 23–60 months), the AOR = 2.31, 95% CI: 1.3, 4.2. Secondly, when IPI cutoff increased by 1 month (IPI<25 vs 25–60 months), the AOR = 3.82, 95%CI: 1.9, 7.8. In both cases, the AORs fall within the reported 95% CI: 1.6, 5.7. Thus, misclassification of IPI, in case it exists, did not affect the observed conclusion even though some difference from the reported estimate (AOR = 2.97) occurs.

The sensitivity analysis was conducted to see whether the conditional logistic regression analysis for the nested case-control study could have affected the estimates and the association when full cohort data (2548 sample) was analyzed by using the classic (unconditional) logistic regression. Accordingly, there is no significant discrepancy between the two approaches. The estimates (odds ratios) are nearly the same: for the nested case-control study (conditional logistic regression), AOR = 2.97 or 3.0, 95% CI (1.6, 5.7), while for the cohort study (unconditional logistic regression), AOR = 3.26 or 3.3, 95% CI (1.9, 5.7) (S1 Table).

## Discussions

This study aimed at elucidating the association between primary PPH and inter-pregnancy interval. Accordingly, primary PPH was associated with IPI <24 months.

In this study, primary PPH was found to be attributed to IPI under 24 months. The finding of this study suggests that preventing pregnancies that are occurring within 24 months after a live birth will contribute to reducing primary PPH. According to the result, about half (49%) of primary PPH in the study population could have been prevented if IPI under 24 months was prevented. This could probably be due to the hypothesis that adequately spacing pregnancies help the uterine wall to recover from the abnormal process of remodeling of endometrial vessels, incomplete healing of uterine scars, hormonal imbalance, and nutritional depletion, and then make it ready for subsequent pregnancy [29, 31]. Besides, when the inter-pregnancy interval is optimal, the uterine muscle will get an adequate tone for contraction after delivery thereby reduce the risk of atonic uterus-related PPH and subsequent risk of maternal death [31]. Increased interval also avoids lactation stress and pregnancy-breastfeeding overlaps that can potentially deplete maternal nutrition (such as folate, iron, and vitamins) via breastfeeding for the baby already born and trans-placental sharing for the fetus in the womb [31, 43]. The result of this study was supported by the studies from Tanzania [27] and Nigeria [28] where shortly spaced pregnancies were identified as risk factors for PPH. However, it is opposed to the latter study conducted in Tanzania [29] which has reported IPI <24 months had no relation with PPH as compared to those 24–60 months. The difference might be explained by a variation like the nature of data obtained, study design, sample size, analysis, and skill of health care providers as well. Data obtained in the Tanzanian study was 15 years of retrospective, from zonal referral hospital (single source) where higher number of referral cases could probably be detected in excess (high detection rate) and the care providers could be hyper-vigilant so that the result might be over-represented the maternal complications as explained by the authors in their limitation. Study design could be the other possible source of variation; we conducted nested case-control from prospective cohort which is more robust to elucidate temporal relationship than retrospective studies. Some potential PPH cases might be managed early so that no more PPH could occur even in the presence of known risk factors. It is also reported that no maternal complications, like PPH, due to short IPI could occur when women are referred early and get appropriate care such as active management of the third stage of labour using utero-tonic drugs [43]. Oxytocin was given for all women, as active management

of the third stage of labour, which can prevent postpartum hemorrhage even if there is a potential for bleeding. Even if IPI was a risk factor for PPH, giving oxytocin might have prevented the occurrence of PPH and the diagnosis becomes no PPH, which might have diluted the association towards null. These conditions might have resulted in the difference between the two studies. The relationship between IPI and PPH is one of the understudied conditions and needs further prospective studies to replicate the findings. We used both matching and stratification in addition to multivariable adjustment to control confounding and bias in our analysis. Nested case-controls studies are efficient with matching and stratification. Therefore, we could not ignore IPI as a risk factor for PPH. Rather we suggest family planning programs and organizations working on it to give due attention for spacing pregnancies. Even in the urban settings where modern contraceptive options are widely accessible, half (50%) of women were not utilizing the services [32]. This may potentially put them at risk of getting pregnant within a short duration after childbirth and subsequent risk of PPH. To benefit women, their families, and the country at large, increasing contraceptive utilization rate via information, education, and communication of the effects of shortly spaced pregnancies need to be underlined.

Primary PPH can significantly be reduced by administering utero-tonic drugs during the third stage of labour for all women since a considerable proportion of women without any historical risk factor may present with PPH. Thus, birth attendants need to check every woman after delivery for potential bleeding before making a discharge from the health facility.

A nested case-control study is usually born from cohort studies. As a result, it has some of the benefits of a cohort study; it resolves temporal relationships that classic case-control studies frequently cannot. The nested case-control study could be improved by applying conditional logistic regression to sparse data or rare outcomes, as matching reduces bias related to confounding. However, a nested case-control study is prone to loss of power and selection bias as it is born from a sub-sample of a cohort study. Power could be improved by increasing the number of controls per case to some extent. Random sampling of controls from the cohort study (risk set) would help to minimize selection bias, particularly when the cohort study is from a defined population [44, 45].

Despite the attempts made to reduce, this study might have the following limitations: firstly, some cases of PPH might remain undiagnosed or misdiagnosed. The reported PPH might be underestimated as it mainly depends on visual estimation. Secondly, to some extent, selection and recall biases might have occurred as some woman might not be included during recruitment and unable to remember the date of the last childbirth. Despite the limitations, the findings of this study can be generalized for urban places in Ethiopia with a similar population and contexts.

## Conclusions

Primary postpartum hemorrhage was associated with inter-pregnancy interval under 24 months, highlighting the need to increase IPI by improving postpartum modern contraceptive utilization in the community. Counseling couples and educating the community at large about the risk of PPH when IPI is under 24 months need to be underlined.

## Supporting information

**S1 Fig. Flow diagram of the cohort study.**
(TIF)

**S1 File. Measures taken against potential sources of bias.**
(DOCX)

**S1 Dataset.**
(DTA)

**S1 Table. Sensitivity analysis using full cohort data, unconditional logistic regression.**
(DOC)

# Acknowledgments

We are very much thankful to study participants, data collectors, supervisors and health workers for their contributions.

# Author Contributions

**Conceptualization:** Belayneh Hamdela Jena, Gashaw Andargie Biks, Yigzaw Kebede Gete, Kassahun Alemu Gelaye.

**Data curation:** Belayneh Hamdela Jena, Gashaw Andargie Biks, Kassahun Alemu Gelaye.

**Formal analysis:** Belayneh Hamdela Jena, Gashaw Andargie Biks, Kassahun Alemu Gelaye.

**Funding acquisition:** Belayneh Hamdela Jena, Gashaw Andargie Biks, Kassahun Alemu Gelaye.

**Investigation:** Belayneh Hamdela Jena, Gashaw Andargie Biks, Yigzaw Kebede Gete, Kassahun Alemu Gelaye.

**Methodology:** Belayneh Hamdela Jena, Gashaw Andargie Biks, Yigzaw Kebede Gete, Kassahun Alemu Gelaye.

**Project administration:** Belayneh Hamdela Jena, Gashaw Andargie Biks, Kassahun Alemu Gelaye.

**Resources:** Belayneh Hamdela Jena, Gashaw Andargie Biks, Kassahun Alemu Gelaye.

**Software:** Belayneh Hamdela Jena, Gashaw Andargie Biks, Kassahun Alemu Gelaye.

**Supervision:** Belayneh Hamdela Jena, Gashaw Andargie Biks, Yigzaw Kebede Gete, Kassahun Alemu Gelaye.

**Validation:** Belayneh Hamdela Jena, Gashaw Andargie Biks, Yigzaw Kebede Gete, Kassahun Alemu Gelaye.

**Visualization:** Belayneh Hamdela Jena, Gashaw Andargie Biks, Kassahun Alemu Gelaye.

**Writing – original draft:** Belayneh Hamdela Jena, Gashaw Andargie Biks, Yigzaw Kebede Gete, Kassahun Alemu Gelaye.

**Writing – review & editing:** Belayneh Hamdela Jena, Gashaw Andargie Biks, Kassahun Alemu Gelaye.

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
