## [Decision Letter · Decision Letter 0]

20 May 2022

PONE-D-21-27673

Association of primary postpartum hemorrhage with inter-pregnancy interval in urban South Ethiopia: a matched nested case control study

PLOS ONE

Dear Dr. Jena,

Thank you for submitting your manuscript to PLOS ONE. After careful consideration, we feel that it has merit but does not fully meet PLOS ONE’s publication criteria as it currently stands. Therefore, we invite you to submit a revised version of the manuscript that addresses the points raised during the review process.

We look forward to receiving your revised manuscript.

Kind regards,

Orvalho Augusto, MD, MPH

Academic Editor

PLOS ONE

“  We would like to thank the University of Gondar and Wachemo University for their financial support. We are very much thankful to study participants, data collectors, supervisors and health workers for their contributions.”

Additional Editor Comments:

This is an interesting study to assess the association between inter-pregnancy interval (IPI) and postpartum haemorrhage (PPH). The authors used a nested matched case-control in a cohort of pregnant recruited after the first trimester in a rural community. They provide a good reason for such an approach (improve statistical efficiency). Usually, this design as well allows the investigators to put extra resources (usually few) in a subsample (in this case use them in 365 rather than in the whole 2546).

Issues:

1. The background is good. However, we still lack a clear description of the potential mechanism the IPI would lead to PPH. And please, provide us with the concept of “primary PPH”.

2. Please put citations for the software used including open-Epi, Epidata, Stata and R. And please write Stata not STATA.

3. Why line 120 says cohorts?

4. Lines 134 and 135 are critical as they define the controls. Please clarify, does this mean only women delivering without PPH; or women still at observation without the outcome?

5. Correct the title. It should be “case-control” not “case control”.

6. The Power/Sample details suggest being a post-hoc power calculation. It is quite strange the authors would know in advance that the OR would be 2.97. Please clarify. If it is a post-hoc power please remove from the text.

- Please a similar concerns by the reviewer below.

7. Line 71 - please correct. Africa and Asia are not countries. Are continents.

8. The bracket opened at the beginning of the line 83 never closes. Please check.

9. Tables 1 and 2 please make sure all percentages have one decimal place.

10. In the limitations. Please move the sensitivity analysis to somewhere in the results.

Reviewers' comments:

Reviewer's Responses to Questions

**Comments to the Author**

1. Is the manuscript technically sound, and do the data support the conclusions?

Reviewer #1: Yes

2. Has the statistical analysis been performed appropriately and rigorously? 

Reviewer #1: Yes

3. Have the authors made all data underlying the findings in their manuscript fully available?

Reviewer #1: No

4. Is the manuscript presented in an intelligible fashion and written in standard English?

Reviewer #1: Yes

5. Review Comments to the Author

Reviewer #1: This manuscript reports a nested case control study looking at the association between interpregnancy interval (IPI) and the risk of postpartum haemorrhage (PPH) in Ethiopia and found that short IPI (<24 months) was associated with an increased chance of primary PPH. The data collection, study design and analysis are appropriate and the paper is reported clearly. I have a few queries and suggestions below:

1. It is not clear why a nested case control approach was used. If a cohort of 2456 pregnant women were recruited and classified on the basis of their most recent IPI, why was a cohort study not undertaken?

2. It seems a little odd that exactly half the cohort had a short IPI. Was this deliberately done at the time of recruitment?

3. Table 3 should present the variables adjusted for as the adjusted odds ratio.

4. "Despite the limitations, nested case control studies are efficient in elucidating temporal relationships, especially when supplemented with matching and stratification (conditional analysis)". This sentence in the discussion should be rephrased. I am not sure that this really qualifies as a nested case control study as the approach was that of a matched case control. Why is matching and stratification superior to adjustment - which has also been done in this analysis?

6. PLOS authors have the option to publish the peer review history of their article (what does this mean?). If published, this will include your full peer review and any attached files.

Reviewer #1: **Yes: **Sohinee Bhattacharya

---

## [Author Response · Author response to Decision Letter 0]

23 May 2022

To: PLOS ONE Journal, Editorial Office

Subject: Submitting a revised version of manuscript and response to Editor and Reviewers.

Ref: Submission ID PONE-D-21-27673

Title of Article: " Association of primary postpartum hemorrhage with inter-pregnancy interval in urban South Ethiopia: a matched nested case control study"

Authors:

Belayneh Hamdela Jena, Gashaw Andargie Biks, Yigzaw Kebede Gete, Kassahun Alemu Gelaye

We would like to thank the Editor for facilitating, commenting and giving the opportunity to revise our manuscript. We are also grateful to reviewers for sharing their views and constructive comments. The comments are very important which will improve the quality of our manuscript. The point-by-point responses for each of the comments and the revised manuscript are provided in the attached documents. 

Regards,

The authors!

Editor’s comments:

Journal Requirements:

Authors’ response: Dear editor, thank you again for your time and contribution. We looked at the PLOS ONE style requirements to edit the whole manuscript and revised accordingly.

Authors’ response: Thank you dear editor, we edited the ‘funding information’. 

3. Thank you for stating the following in the Acknowledgments Section of your manuscript: “We would like to thank the University of Gondar and Wachemo University for their financial support. We are very much thankful to study participants, data collectors, supervisors and health workers for their contributions.” 

Authors’ response: Dear editor, we have revised the financial disclosure statements. We revised as “The authors received no specific funding for this work.” 

In addition, we removed funding-related text from the manuscript.

Authors’ response: Dear editor, we have uploaded dataset as supporting information (S1 Dataset).

Authors’ response: Thank you dear editor, we provided data set as supporting information (S1 Dataset). We stated in online system as “All relevant data are within the manuscript and its Supporting Information files”.

Authors’ response: Dear editor, thank you we put ethics statement in the methods section.

Additional Editor Comments:

This is an interesting study to assess the association between inter-pregnancy interval (IPI) and postpartum haemorrhage (PPH). The authors used a nested matched case-control in a cohort of pregnant recruited after the first trimester in a rural community. They provide a good reason for such an approach (improve statistical efficiency). Usually, this design as well allows the investigators to put extra resources (usually few) in a subsample (in this case use them in 365 rather than in the whole 2546).

Issues:

1. The background is good. However, we still lack a clear description of the potential mechanism the IPI would lead to PPH. And please, provide us with the concept of “primary PPH”.

Authors’ response: Thank you dear editor for your time and contribution to this manuscript. Your inputs will improve the quality of our manuscript. We acknowledge your comments. In the revised manuscript we added 1 paragraph about potential causal mechanisms. As we defined early in the paragraph, and subsequent paragraphs of the introduction section, primary PPH is excess bleeding that occurs within 24 hours after delivery. Primary refers to the time (within 24 hours). It is commonly due to direct causes such as uterine atony, retained placental tissues, uterine rupture/lacerations, etc, and the main cause of maternal death, as we reported in the introduction. If bleeding occurs after 24 hours but before 12 weeks after delivery, then it is defined as secondary PPH. Secondary PPH is usually secondary to puerperal sepsis/infection and other causes. Our focus was on primary due to its impact and feasibility of data on the diagnosis.

2. Please put citations for the software used including open-Epi, Epidata, Stata and R. And please write Stata not STATA.

Authors’ response: Dear editor, thank you! We put citations for each software inside the text of the revised manuscript, and we showed in the highlighted-text. Note: when removing post-hoc power analysis we also removed Stata, as you commented on comment number 6 below.

3. Why line 120 says cohorts?

Authors’ response: Thank you dear editor, it was editorial problem. Now we corrected it, as ‘a cohort’. 

4. Lines 134 and 135 are critical as they define the controls. Please clarify, does this mean only women delivering without PPH; or women still at observation without the outcome?

Authors’ response: Thank you dear editor. We mean that ‘women delivering without PPH’. It did not include those who were lost of follow-up, and abortion cases. We corrected it in the text. 

5. Correct the title. It should be “case-control” not “case control”.

Authors’ response: Thank you dear editor, we corrected as “case-control” throughout the manuscript.

6. The Power/Sample details suggest being a post-hoc power calculation. It is quite strange the authors would know in advance that the OR would be 2.97. Please clarify. If it is a post-hoc power please remove from the text.

- Please a similar concerns by the reviewer below.

Authors’ response: Thank you dear editor for the comment. We acknowledge your view. It was pot-hoc power, which was just to estimate whether the sample size was adequate or not. Now we removed it. 

7. Line 71 - please correct. Africa and Asia are not countries. Are continents.

Authors’ response: thank you dear editor, we corrected as ‘continents’.

8. The bracket opened at the beginning of the line 83 never closes. Please check.

Authors’ response: Dear editor, we checked it, and closed the bracket.

9. Tables 1 and 2 please make sure all percentages have one decimal place.

Authors’ response: Dear editor, we checked for it, and all have only one decimal, except P-values in Table 1 and 2. Thank you!

10. In the limitations. Please move the sensitivity analysis to somewhere in the results.

Authors’ response: Thank you dear editor, now we moved the sensitivity analysis in the result section under the regression table. Just before, discussion section. Thank you for the input.

Reviewers’ comments

Reviewer #1

This manuscript reports a nested case control study looking at the association between interpregnancy interval (IPI) and the risk of postpartum haemorrhage (PPH) in Ethiopia and found that short IPI (<24 months) was associated with an increased chance of primary PPH. The data collection, study design and analysis are appropriate and the paper is reported clearly. I have a few queries and suggestions below: 

1. It is not clear why a nested case control approach was used. If a cohort of 2456 pregnant women were recruited and classified on the basis of their most recent IPI, why was a cohort study not undertaken?

Authors’ response: Thank dear reviewer for your time and contribution to this manuscript. Your comments will improve the quality of our manuscript. We acknowledge your views. We considered a nested case-control study for the following reasons: we faced the outcome (primary PPH) was rare (for statistical purpose). In this case, case-control study is efficient (for rare case). We want to apply more robust statistical methods (matched and paired analysis). We did a matched analysis to increase the strength of analysis for more valid estimate. Matching for known confounder (e.g. age) in addition to adjustment for other risk factors in multivariable model increase statistical efficiency. Matching for a variable in the whole cohort is not feasible. In addition, matching for ‘location (kebele)’ makes some unobserved sources of variation more similar such as neighborhood and health service related characteristics. Kindly, see additional reasons in your comment number 4 below!

2. It seems a little odd that exactly half the cohort had a short IPI. Was this deliberately done at the time of recruitment?

Authors’ response: Thank you dear reviewer for this crucial comment. At the beginning of the study, our thinking was just to take equal number of exposed and unexposed groups. While completing the enrollment, we randomly excluded 32 individuals (as we showed in Sl flow diagram in previous submission) from the unexposed group just to equalize the sample size for both group. A similar comment as you was given by other reviewer in one of our study related to this (https://doi.org/10.1186/s12884-021-04325-z ), and the reviewer suggest us to go back to collect the outcomes status of those 32 individuals data dropped. Later, we the authors discussed, and decided to include the outcome data of the 32 participant by going back to the health facilities since we already had a baseline data at hand together with their addresses. So we trace back and collected the outcome data to include in the analysis. Thereby we also collected the data for the primary PPH too. Therefore, in this revised manuscript we added the information about those 32 participants in the sample size, flow diagram and figure as well. Of course, we could not get a woman diagnosed for primary PPH among those 32 participants. We also checked if they should be matched for age and location (kebele). However, the data of the 32 participants were not eligible for the matching and stratification. Thus, we did not include them in the analysis of the nested case-control study. Rather we report in sample size sections, figure and flow diagram in the revised manuscript. Dear reviewer as you really commented, and the same comment from the other reviewer in related article, it is not mandatory to take equal number of exposed and unexposed group for cohort study. Thank you again for your input.

3. Table 3 should present the variables adjusted for as the adjusted odds ratio.

Authors’ response: Thank you dear reviewer, we revised Table 3. Kindly, see the table 3 in the text in the revised manuscript.

4. "Despite the limitations, nested case control studies are efficient in elucidating temporal relationships, especially when supplemented with matching and stratification (conditional analysis)". This sentence in the discussion should be rephrased. I am not sure that this really qualifies as a nested case control study as the approach was that of a matched case control. Why is matching and stratification superior to adjustment - which has also been done in this analysis?

Authors’ response: Thank you dear reviewer, we removed it from the discussion as it was repeatedly used. Of course, we applied both matching & stratification and adjustment as well. Both matching and adjustment are used to control confounding. We are not making matching and stratification superior to adjustment. We matched for “age” as it is a well-known biological confounder in many of health-related outcomes including PPH. Matching for location makes cases and controls more similar in terms of neighborhood circumstances, so that the cases and controls represent the population from where they come from. For sparse data, data from different geographic location, and rare conditions, nested case-control study via applying conditional analysis (matched and paired analysis) is appropriate. Using both matching and adjustment increase statistical strength.

END!

---

## [Decision Letter · Decision Letter 1]

7 Jun 2022

PONE-D-21-27673R1Association of primary postpartum hemorrhage with inter-pregnancy interval in urban South Ethiopia: a matched nested case-control studyPLOS ONE

Dear Dr. Jena,

Thank you for submitting your manuscript to PLOS ONE. After careful consideration, we feel that it has merit but does not fully meet PLOS ONE’s publication criteria as it currently stands. Therefore, we invite you to submit a revised version of the manuscript that addresses the points raised during the review process.

We look forward to receiving your revised manuscript.

Kind regards,

Orvalho Augusto, MD, MPH

Academic Editor

PLOS ONE

Journal Requirements:

Additional Editor Comments (if provided):

Thank you for responding to all our comments and questions.

The nested case-control design is not yet a commonly known design. This design is born within a cohort from which cases are identified. Then matched controls are found from the cohort by a density sampling (your case here) or baseline sampling.

The main advantage is that extra (and usually expensive measurements like, for example, biomarkers) information can be collected for this relatively small sample. In particular, the density sampling in a case-control can help approximate a relative-risk without the need for the low risk/incidence condition in the overall population. However, the case-control still potentially suffers from selection bias and we have to agree with the reviewer there is a lot calculations involved to analyse this.

Therefore, I would suggest:

1) Please, do the extra analysis (if data is available as the reviewer recommends)

2) Comment about those results and put those tables in the Supplementary materials.

3) Please, add in the strengths and limitations a discussion about this particular design.

Reviewers' comments:

Reviewer's Responses to Questions

**Comments to the Author**

1. If the authors have adequately addressed your comments raised in a previous round of review and you feel that this manuscript is now acceptable for publication, you may indicate that here to bypass the “Comments to the Author” section, enter your conflict of interest statement in the “Confidential to Editor” section, and submit your "Accept" recommendation.

Reviewer #1: All comments have been addressed

2. Is the manuscript technically sound, and do the data support the conclusions?

Reviewer #1: Partly

3. Has the statistical analysis been performed appropriately and rigorously? 

Reviewer #1: Yes

4. Have the authors made all data underlying the findings in their manuscript fully available?

Reviewer #1: No

5. Is the manuscript presented in an intelligible fashion and written in standard English?

Reviewer #1: Yes

6. Review Comments to the Author

Reviewer #1: I am still not clear why matching, stratification and adjustment was required for accounting for confounding. I would urge the authors to conduct logistic regression analysis adjusting for all potential confounders at least as a sensitivity analysis including all women recruited in the cohort. I am a great advocate for keeping things simple and it always worries me when complicated statistical analysis is done without any substantial gain.

7. PLOS authors have the option to publish the peer review history of their article (what does this mean?). If published, this will include your full peer review and any attached files.

Reviewer #1: **Yes: **Sohinee Bhattacharya

---

## [Author Response · Author response to Decision Letter 1]

9 Jun 2022

To: PLOS ONE Journal, Editorial Office

Subject: Submitting a revised version of manuscript and response to Editor and Reviewers.

Ref: Submission ID PONE-D-21-27673R1

Title of Article: " Association of primary postpartum hemorrhage with inter-pregnancy interval in urban South Ethiopia: a matched nested case control study"

Authors:

Belayneh Hamdela Jena, Gashaw Andargie Biks, Yigzaw Kebede Gete, Kassahun Alemu Gelaye

We would like to thank the Editor for facilitating, commenting and giving the opportunity to revise our manuscript. We are also grateful to reviewers for sharing their views and constructive comments. The comments are very important which will improve the quality of our manuscript. The point-by-point responses for each of the comments and the revised manuscript are provided in the attached documents. 

Regards,

The authors!

Editor’s comments:

Journal Requirements:

Authors’ response: Thank you dear editor for your time, immense contribution, and giving the opportunity to revise the manuscript. We acknowledge your constructive comments, and suggestions, which will improve the quality of our manuscript indeed. We have checked the references for any retraction, and no retracted references found.

Additional Editor Comments (if provided):

Thank you for responding to all our comments and questions.

The nested case-control design is not yet a commonly known design. This design is born within a cohort from which cases are identified. Then matched controls are found from the cohort by a density sampling (your case here) or baseline sampling.

The main advantage is that extra (and usually expensive measurements like, for example, biomarkers) information can be collected for this relatively small sample. In particular, the density sampling in a case-control can help approximate a relative-risk without the need for the low risk/incidence condition in the overall population. However, the case-control still potentially suffers from selection bias and we have to agree with the reviewer there is a lot calculations involved to analyse this.

Therefore, I would suggest:

1) Please, do the extra analysis (if data is available as the reviewer recommends)

Authors’ response: Thank you dear editor, we acknowledge your suggestion, and reviewer’s as well. We already have full data for the participants. Primary postpartum hemorrhage (PPH) was one of the outcomes hypothesized to have association with inter-pregnancy interval during designing the study. Thus, we have conducted the sensitivity analysis using 2548 sample data for which the outcome was ascertained, as suggested by reviewer#1, using logistic regression. The result of the sensitivity analysis (full data containing the regression analysis) was presented in supplementary table 1 (S1 Table). 

We have included the result of the sensitivity analysis in the result section of the revised manuscript, under sensitivity analysis section, and supplementary table was also cited.

2) Comment about those results and put those tables in the Supplementary materials.

Authors’ response: Dear editor, as you can see from the supplementary table 1, no significant difference on the conclusion and the direction of association was observed between the two approaches (nested case-control study and cohort study), except very few difference on the estimates; that means AOR=2.97~3.0, 95% CI (1.6, 5.7) for the nested case-control study while AOR=3.26 ~3.3, 95% CI (1.9, 5.7) for the cohort study. In both study designs, the results are nearly the same, and the interpretation is the same: the odds of primary postpartum hemorrhage was 3 times higher for women with inter-pregnancy interval under 24 months as compared to women with inter-pregnancy interval 24-60 months. Since the data on primary postpartum hemorrhage was rare, and sparse because the primary PPH was from five urban settings, which are geographically diverse. For such sparse data, nested case-controls studies highly appropriate. Nested case-controls study is efficient with matching for known biological confounders such as age. Such matching, when needed, is not feasible for large sample of cohort, like ours 2548, when the outcome is rare or sparse. As you really mentioned above, nested case-control study is primarily relevant for rare conditions like biomarkers, it is also commonly being applicable for disease, death and behavioral conditions. Kindly, I have included just few literature (of course many) on nested case-control studies regarding the areas of application. 

- https://doi.org/10.1186/s12889-018-5757-2. 

- DOI:10.1371/journal.pone.0159390.

- DOIhttps://doi.org/10.1186/s12879-017-2933-4. 

- https://doi.org/10.3748/wjg.v9.i1.99. 

- https://doi.org/10.1038/ki.2014.74

3) Please, add in the strengths and limitations a discussion about this particular design.

Authors’ response: Thank you dear editor, we have added one paragraph about the strengths and limitations of nested case-control study, including some references.

Reviewers’ comments

Reviewer #1

I am still not clear why matching, stratification and adjustment was required for accounting for confounding. I would urge the authors to conduct logistic regression analysis adjusting for all potential confounders at least as a sensitivity analysis including all women recruited in the cohort. I am a great advocate for keeping things simple and it always worries me when complicated statistical analysis is done without any substantial gain.

Authors’ response: Thank you again for your time and immense contribution. We acknowledge your comments, which will improve the quality of our manuscript. The main reason for matching is to control confounding like age, and matching in location reduces geographic disparities for such sparse data. Matched analysis (conditional logistic regression) is highly appropriate to manage sparse data problem or sparse data bias. 

We also appreciate your suggestion to conduct sensitivity analysis to see whether the conditional logistic regression analysis for nested case-control study could have affected the estimates and the association when full cohort data was analyzed by using the classic (unconditional) logistic regression. According to the sensitivity analysis (supplementary table 1), there is no much discrepancy between the two approaches. The estimates (odds ratios) are nearly the same: for the nested case-control study (conditional logistic regression) AOR=2.97~3.0, 95% CI (1.6, 5.7) while for the cohort study (unconditional logistic regression) AOR=3.26 ~3.3, 95% CI (1.9, 5.7). Since both approaches estimated nearly similar estimates the interpretations are the same; the odds of primary postpartum hemorrhage was 3 times higher for women with inter-pregnancy interval under 24 months as compared to women with inter-pregnancy interval 24-60 months. 

END!

---

## [Decision Letter · Decision Letter 2]

27 Jun 2022

Association of primary postpartum hemorrhage with inter-pregnancy interval in urban South Ethiopia: a matched nested case-control study

PONE-D-21-27673R2

Dear Dr. Jena,

We’re pleased to inform you that your manuscript has been judged scientifically suitable for publication and will be formally accepted for publication once it meets all outstanding technical requirements.

Kind regards,

Orvalho Augusto, MD, MPH

Academic Editor

PLOS ONE

Additional Editor Comments (optional):

Just few outstanding issues:

- Tables 1 and 2 - Please remove the total column and the Chi-squared statistic (just leave the p-value).

Reviewers' comments:

Reviewer's Responses to Questions

**Comments to the Author**

1. If the authors have adequately addressed your comments raised in a previous round of review and you feel that this manuscript is now acceptable for publication, you may indicate that here to bypass the “Comments to the Author” section, enter your conflict of interest statement in the “Confidential to Editor” section, and submit your "Accept" recommendation.

Reviewer #1: All comments have been addressed

2. Is the manuscript technically sound, and do the data support the conclusions?

Reviewer #1: Yes

3. Has the statistical analysis been performed appropriately and rigorously? 

Reviewer #1: Yes

4. Have the authors made all data underlying the findings in their manuscript fully available?

Reviewer #1: No

5. Is the manuscript presented in an intelligible fashion and written in standard English?

Reviewer #1: Yes

6. Review Comments to the Author

Reviewer #1: Thank you for addressing all comments and suggestions. The manuscript now reads better and should be acceptable for publication.

7. PLOS authors have the option to publish the peer review history of their article (what does this mean?). If published, this will include your full peer review and any attached files.

Reviewer #1: **Yes: **Sohinee Bhattacharya

---

## [Editor Report · Acceptance letter]

8 Jul 2022

PONE-D-21-27673R2 

Association of primary postpartum hemorrhage with inter-pregnancy interval in urban South Ethiopia: a matched nested case-control study 

Dear Dr. Jena:

I'm pleased to inform you that your manuscript has been deemed suitable for publication in PLOS ONE. Congratulations! Your manuscript is now with our production department. 

Kind regards, 

on behalf of

Dr. Orvalho Augusto 

Academic Editor

PLOS ONE